# Best Practices for Spatial Profiling for Breast Cancer Research with the GeoMx^®^ Digital Spatial Profiler

**DOI:** 10.3390/cancers13174456

**Published:** 2021-09-04

**Authors:** Helga Bergholtz, Jodi M. Carter, Alessandra Cesano, Maggie Chon U Cheang, Sarah E. Church, Prajan Divakar, Christopher A. Fuhrman, Shom Goel, Jingjing Gong, Jennifer L. Guerriero, Margaret L. Hoang, E. Shelley Hwang, Hellen Kuasne, Jinho Lee, Yan Liang, Elizabeth A. Mittendorf, Jessica Perez, Aleix Prat, Lajos Pusztai, Jason W. Reeves, Yasser Riazalhosseini, Jennifer K. Richer, Özgür Sahin, Hiromi Sato, Ilana Schlam, Therese Sørlie, Daniel G. Stover, Sandra M. Swain, Alexander Swarbrick, E. Aubrey Thompson, Sara M. Tolaney, Sarah E. Warren

**Affiliations:** 1Department of Cancer Genetics, Institute for Cancer Research, Oslo University Hospital, 0450 Oslo, Norway; Helga.Bergholtz@rr-research.no (H.B.); therese.sorlie@rr-research.no (T.S.); 2Department of Laboratory Medicine and Pathology, Mayo Clinic, Rochester, MN 55905, USA; Carter.Jodi@mayo.edu; 3ESSA Pharma Inc., South San Francisco, CA 94080, USA; acesano@essapharma.com; 4ICR Clinical Trials and Statistics Unit, Division of Clinical Studies, The Institute of Cancer Research, London SM2 5NG, UK; maggie.cheang@icr.ac.uk; 5NanoString® Technologies Inc., Seattle, WA 98109, USA; schurch@nanostring.com (S.E.C.); pdivakar@nanostring.com (P.D.); kfuhrman@nanostring.com (C.A.F.); jgong@nanostring.com (J.G.); mhoang@nanostring.com (M.L.H.); yliang@nanostring.com (Y.L.); Jessie.perez@curibio.com (J.P.); jreeves@nanostring.com (J.W.R.); hsato@nanostring.com (H.S.); 6Peter MacCallum Cancer Centre, Melbourne, VIC 3000, Australia; Shom.goel@petermac.org; 7Sir Peter MacCallum Department of Oncology, University of Melbourne, Parkville, VIC 3010, Australia; 8Division of Breast Surgery, Department of Surgery, Brigham and Women’s Hospital, Boston, MA 02115, USA; jguerriero@bwh.harvard.edu (J.L.G.); emittendorf@bwh.harvard.edu (E.A.M.); 9Duke Cancer Institute, Duke University, Durham, NC 27710, USA; shelley.hwang@duke.edu; 10Rosalind and Morris Goodman Cancer Centre, McGill University, Montreal, QC H3A 0G4, Canada; hellen.kuasne@mcgill.ca; 11Knight Cancer Institute, Oregon Health and Science University, Portland, OR 97239, USA; leejin@ohsu.edu; 12Breast Oncology Program, Dana-Farber Cancer Institute, Boston, MA 02215, USA; 13Harvard Medical School, Boston, MA 02115, USA; Sara_Tolaney@DFCI.HARVARD.EDU; 14Translational Genomics and Targeted Therapies in Solid Tumors, August Pi i Sunyer Biomedical Research Institute, 08036 Barcelona, Spain; alprat@clinic.cat; 15Yale Cancer Center, Yale School of Medicine, New Haven, CT 06510, USA; lajos.pusztai@yale.edu; 16Department of Human Genetics, McGill University, Montreal, QC H3A 0G4, Canada; yasser.riazalhosseini@mcgill.ca; 17McGill University Genome Centre, McGill University, Montreal, QC H3A 0G4, Canada; 18Department of Pathology, University of Colorado Anschutz Medical Campus, Aurora, CO 80045, USA; jennifer.richer@cuanschutz.edu; 19Department of Drug Discovery and Biomedical Sciences, University of South Carolina, Columbia, SC 29208, USA; sahin@cop.sc.edu; 20MedStar Washington Hospital Center, Washington, DC 20010, USA; schlamilana@gmail.com; 21Tufts Medical Center, Boston, MA 02111, USA; 22Institute of Clinical Medicine, University of Oslo, 0315 Oslo, Norway; 23Ohio State University Comprehensive Cancer Center, Columbus, OH 43210, USA; Daniel.Stover@osumc.edu; 24Georgetown Lombardi Comprehensive Cancer Center, Washington, DC 20057, USA; sandra.swain@georgetown.edu; 25Georgetown University Medical Center, Washington, DC 20057, USA; 26MedStar Health, Washington, DC 20057, USA; 27Garvan Institute of Medical Research, Darlinghurst, NSW 2010, Australia; a.swarbrick@garvan.org.au; 28St Vincent’s Clinical School, Faculty of Medicine, UNSW Sydney, Sydney NSW 2052, Australia; 29Department of Cancer Biology, Mayo Clinic Florida, Jacksonville, FL 32224, USA; Thompson.Aubrey@mayo.edu; 30Department of Medical Oncology, Dana-Farber Cancer Institute, Boston, MA 02115, USA

**Keywords:** breast cancer, spatial biology, RNA and protein profiling, GeoMx, digital spatial profiler, tumor microenvironment, biomarker discovery, whole transcriptome atlas, cancer transcriptome atlas, tumor heterogeneity

## Abstract

**Simple Summary:**

In breast cancer, there is a high degree of variability in tumors and the surrounding tissue called the tumor microenvironment (TME). To better understand tumor biology and metastasis, as well as to predict response to cancer treatments or the course of the disease, it is important to characterize molecular diversity in the breast TME. The GeoMx Digital Spatial Profiler (DSP) enables researchers to spatially analyze proteins and RNA transcripts in tumors and surrounding tissues from patients or preclinical models. Using the GeoMx DSP, protein expression and RNA transcripts in the distinct regions of a tumor can be quantified up to and including the whole transcriptome level. Herein, the GeoMx Breast Cancer Consortium presents best practices for GeoMx spatial profiling of tumors to promote the collection of high-quality data, optimization of data analysis and integration of datasets to accelerate biomarker discovery. These best practices can also be applied to any tumor type to provide information about the tumor and the TME.

**Abstract:**

Breast cancer is a heterogenous disease with variability in tumor cells and in the surrounding tumor microenvironment (TME). Understanding the molecular diversity in breast cancer is critical for improving prediction of therapeutic response and prognostication. High-plex spatial profiling of tumors enables characterization of heterogeneity in the breast TME, which can holistically illuminate the biology of tumor growth, dissemination and, ultimately, response to therapy. The GeoMx Digital Spatial Profiler (DSP) enables researchers to spatially resolve and quantify proteins and RNA transcripts from tissue sections. The platform is compatible with both formalin-fixed paraffin-embedded and frozen tissues. RNA profiling was developed at the whole transcriptome level for human and mouse samples and protein profiling of 100-plex for human samples. Tissue can be optically segmented for analysis of regions of interest or cell populations to study biology-directed tissue characterization. The GeoMx Breast Cancer Consortium (GBCC) is composed of breast cancer researchers who are developing innovative approaches for spatial profiling to accelerate biomarker discovery. Here, the GBCC presents best practices for GeoMx profiling to promote the collection of high-quality data, optimization of data analysis and integration of datasets to advance collaboration and meta-analyses. Although the capabilities of the platform are presented in the context of breast cancer research, they can be generalized to a variety of other tumor types that are characterized by high heterogeneity.

## 1. Introduction

### 1.1. Biology and Prognostic Biomarkers in Breast Cancer 

In recent years, improvements in diagnostic tools and treatments have led to better outcomes for patients with breast cancer; however, there are still subgroups of patients with aggressive disease where effective therapies remain elusive [1,2]. Breast cancer is a heterogenous disease, both biologically and clinically, with a high degree of inter- and intra-patient variability that is intrinsic to the tumor cells and in the cellular and molecular composition of the tumor microenvironment (TME) [3,4,5,6,7,8]. Clinically, the diversity manifests in patients with respect to response to therapy and prognosis [3,9,10,11,12,13]. 

Within an individual tumor, heterogeneity occurs at the level of the genome, transcriptome and proteome, which leads to the diversity of cell populations and tissue morphology [3,9,14,15]. As an example, recent studies have shown that tumor-infiltrating lymphocytes (TILs) in the primary tumor play an important role in therapy response and prognosis in breast cancer [4,16,17,18,19,20,21,22,23,24,25,26]. Yet, improved clinical outcomes were associated with the presence of TILs in some but not all tumor types and in some but not all patients [4,27,28,29,30]. Conventional diagnostic tools, including immunohistochemistry (IHC), bulk DNA sequencing, or RNA expression analysis, do not measure intra-patient and intra-tumor heterogeneity to its full extent, either because of limited plex (e.g., IHC) or lack of spatial resolution (e.g., bulk analysis of nucleic acids) [9,31,32,33,34]. Therefore, an opportunity exists to improve available tools for characterizing and quantifying the molecular heterogeneity of breast tumors (as well as other highly histologically and clinically heterogeneous tumors) and associating those signatures with clinical outcomes to improve the performance of prognostic and predictive biomarkers [35,36]. 

### 1.2. Opportunity of Spatial Biology in Breast Cancer

Tumors are not uniform masses of malignant cells, but rather collections of multi-faceted local ecosystems characterized by cellular heterogeneity, dynamicity and plasticity, as well as complex cell–cell interactions. Different layers of heterogeneity in the TME can be recognized as (1) genetic and epigenetic features of cancer cells (e.g., somatic mutations, copy-number variations, differences in the epigenome and proteome), (2) non-malignant cells (e.g., immune cells, fibroblasts), (3) different extracellular matrix (ECM) composition and degree of remodeling and (4) cellular cross-talk as a result of expressed/secreted molecules, antigenic display and receptor diversity. Heterogeneity can arise as a result of tumor evolution or the selective pressure of treatment and drug resistance within a single site or as the tumor metastasizes throughout the body. It can be investigated by profiling regions within a single tumor, comparing paired samples of primary tumors to metastases, comparing serial biopsies pre- versus post-treatment, comparing paired samples of drugs sensitive to resistant tumors, or investigating patient-to-patient clinical and biological variability [3,37,38]. 

Conventional technologies, either high-plex bulk genetic/transcriptional profiling (e.g., genotyping, microarray, RNA-seq, nCounter^®^, or mass spectrometry) or low-plex, spatially resolved profiling data (e.g., fluorescence in situ hybridization or IHC), are technically limited by the lack of association between the capability of high-plex molecular profiling and the spatial resolution. These limitations prevent a holistic and integrated understanding of the tumor biology that plays a role in tumor growth, dissemination, response and/or emerging adaptation to therapies. Recent technological advances in a high-plex spatial profiling platform [39,40] open the door to enable characterization of the different layers of heterogeneity in the TME, e.g., features of neoplastic cells, immune cells, stromal fibroblasts and cellular cross-talk. Molecular spatial profiling can therefore uniquely capture snapshots of cellular heterogeneity, as well as the phenotypic or functional evolution occurring within different areas of the tumor and TME, which could be highly relevant from a prognostic/predictive point of view. 

### 1.3. GeoMx Digital Spatial Profiler (DSP) for Spatially Resolved Biology

The GeoMx DSP was developed to enable spatially resolved, high-plexed molecular profiling of tissues with digital quantitation of target analytes [41,42]. Researchers can quantify proteins or RNA transcripts using barcoded DNA oligos attached to primary antibodies (for protein) or in situ hybridization probes (for RNA) via an ultraviolet (UV)-photocleavable linker (Figure 1). These detection reagents are applied to the surface of the tissue in parallel with up to four customizable fluorescent morphology markers for visualizing the tissue architecture [43]. The sample is imaged and regions of interest (ROIs) are exposed to UV light that cleaves the linker and releases the barcoded oligos for capture by microfluidics and off-instrument quantitation via nCounter or next generation sequencing (NGS) (Figure 1). To date, more than 300 antibodies have been validated for the GeoMx platform and RNA profiling has been developed to the whole transcriptome level for human and mouse samples and protein profiling of 100-plex for human samples. Both protein and RNA content can be customized with additional targets as needed. 

The platform enables a high degree of customization to profile ROIs of any shape within the viewing area, from the minimum region size of 5 µm × 5 µm, up to a maximum size of 660 µm × 785 µm. Additionally, UV-light can be masked to focus on the fluorescent signal of individual morphology marker in a given ROI, subdividing that ROI into distinct cellular compartments called areas of illumination (AOI) that can be profiled and readout separately. For example, the use of the epithelial cell-specific marker pan-cytokeratin (PanCK) assists pathologic identification of breast tumor tissue within a sample. Each PanCK-positive or PanCK-negative compartment can be profiled by separately illuminating the compartment of interest with UV light to release the oligo barcodes. Similarly, cell-type-specific markers (e.g., CD3 on T cells) can be used to profile those specific cell populations within an ROI, without the need for these cells to be contiguous. Complex mask can be generated for cellular compartments defined by the presence or absence of multiple morphology markers (e.g., CD68-positive and CD11b-positive macrophages).

The GeoMx DSP platform has a number of advantages for spatial profiling. First, it is compatible with both formalin-fixed paraffin-embedded (FFPE) and frozen tissues [44]. In addition, it is suited for a variety of sample sizes, from core-needle biopsies (CNBs) and tissue microarray (TMA) to tissue sections from resected tumors. Moreover, it only requires a single ~5 µm tissue section and the same section can be repeatedly profiled due to the relatively non-destructive process of GeoMx DSP. As described, it is capable of high-plex spatial profiling based on highly reproducible digital counts of RNA and protein [42]. Multiple configurations of pre-designed and pre-validated reagents have been developed and custom antibody-based or RNA probe-based reagents to targets, such as endogenous or exogenous proteins, non-coding RNA species and exon-level profiling of transcribed RNAs, can be included with the GeoMx reagent panels. In addition, custom protein-based or RNA-based morphology markers (e.g., mRNA in situ hybridization via RNAscope) for ROI selection and segmentation can be tailored to the researcher’s experimental approach. The ROI profiling strategies can be customized for the specific spatial compartments for profiling. With these key advantages, GeoMx allows researchers to achieve a wide spectrum of spatially resolved transcriptomic and proteomic characterizations of breast cancer samples [4,35,45,46,47]. Furthermore, the GeoMx spatial profiling is applicable for other tumor types, such as melanoma, lung, prostate, or liver cancers, characterized by high clinical and histological heterogeneity [48,49,50,51,52,53,54,55,56,57,58,59,60]. 

The GeoMx Breast Cancer Consortium (GBCC) is composed of breast cancer researchers who are developing and promoting innovative approaches for spatial profiling to facilitate progress in novel biomarker discovery. GBCC members have committed to sharing data and experimental design strategies to accelerate scientific progress in breast cancer research. Here, we present the current best practices from the GBCC for GeoMx profiling to promote acquisition of the highest quality data and facilitate data comparability and integration. 

## 2. Defining the Questions in Breast Cancer Biology

Prior to employing any technology to understand cancer biology, it is helpful to know what questions the technology is best suited to answer. Here, we provide a list of questions that spatial biology and the GeoMx platform are uniquely poised to address, using the framework of tumor evolution as an example. 

Tumors evolve due to intrinsic mutations of tumor cells and in response to outside stimuli, such as the pressures of immune selection, the impact of the TME and adaptations to treatment regimens [61,62]. Evolution can be sped up or shaped by therapeutic pressure (e.g., outgrowth of resistant mutant clones) or slowed by surgical debulking, which reduces the total diversity [61,62]. Cancer evolution impacts molecular and morphological tumor heterogeneity and complicates the design of effective interventions. To address this issue, profiling primary tumors and recurrent tumors, primary tumors and metastases, or paired samples pre-, during and post-treatment enables a more comprehensive characterization of tumor evolution. 

Clinical heterogeneity in response to therapy can be observed within an individual patient, i.e., a particular treatment may lead to a mixed response [63]. It can also be observed between patients, where histologically similar tumors show clinical response to a given therapy in one patient, while progression in another patient [64,65]. In both cases, the driving factors of the clinical heterogeneity are often unclear. GeoMx and other spatial biology approaches may be able to answer these clinical questions using customizable analytical strategies such as (1) profiling across location, (2) profiling across time, particularly in trials where serial biopsies are collected pre-, during and post-treatment and/or from primary tumor and metastases from multiple sites, and (3) profiling across preclinical and clinical samples. 

### 2.1. Profiling across Location

Within a single tumor, heterogeneity of expression patterns can be induced by differences in the local microenvironment. For example, the center of the tumor is often hypoxic with low pH, where cells become necrotic, releasing factors associated with cell death or changing the expression of immunomodulatory genes [66]. Similarly, highly vascularized areas of the tumor have different nutrient gradients than areas that have less exposure to the bloodstream. In addition, heterogeneity of tumor cell populations, their expression of driver mutations or drug targets, intra-tumor immune cell populations and abundance of other non-malignant cells in the TME may impact metastatic potential, resistance to therapy and immunogenicity [3]. 

Profiling tumors across multiple anatomical locations, intra-patient but inter-lesion, can provide insight into the evolution of the tumor. When metastasized, tumor cells are exposed to vastly different tissue microenvironments, such as in the brain, where they may be protected by the blood–brain barrier, or the lung, where the oxygen level is high. These environmental conditions may drive the tumor evolution and shape response to treatment [40]. Furthermore, it is important to characterize inter-patient heterogeneity that is associated with differences in clinical outcomes between patients. In breast cancer, age and parity status are examples of patient conditions that can strongly influence the TME at the primary site, but also at metastatic sites, such as the liver, that undergo major pregnancy-induced changes [67,68]. Of course, age and consequent hormonal milieu affect the fitness of both tumor and normal cells at the primary and metastatic sites and the function of immune cells. Many biobanks have amassed large collections of tumor biopsies with associated clinical information that can be highly informative when profiled systematically. Because of these complexities, an assessment of heterogeneity requires specific tools and experimental designs for quantitative and spatial analysis of different putative mechanisms. 

### 2.2. Profiling across Time

Time represents an orthogonal axis one can consider for profiling heterogeneity; time course studies enable profiling prior to and following a therapeutic intervention. Neoadjuvant therapies have become a more common approach, particularly in epidermal growth factor receptor 2 (HER2)-positive, triple-negative breast cancer (TNBC) and large estrogen receptor-positive tumors. This “window of opportunity” design enables tumor profiling before and after drug treatment by profiling the diagnostic biopsy and tissue resected during surgery. One challenge is that the diagnostic sample is often a core-needle biopsy, where the tissue is limited and subjected to sampling “errors”, compared to the excisional tissue that retains more information regarding tissue organization. 

Additionally, when neoadjuvant therapy is effective, tumor cellularity profoundly changes between pre- and post-treatment leading to a significant challenge in data interpretation at bulk RNA level [15]. To overcome this challenge, a two-pronged analytical strategy should be employed. One analysis should focus only on the pre-treatment samples to look for biomarkers associated with response to therapy. The other analysis should be a patient- and cohort-level analysis of changes induced by treatment where residual tumor remains in the post-treatment sample. To capture treatment-induced changes in the cancer cells, tumor-cell-specific ROIs can be selected by GeoMx. Spatial profiling can provide insights into the pharmacodynamic (PD) effects of a drug, as well as possible emergent compensatory changes and clonal selection that may occur in response to the therapeutic selection. Similar profiling strategies can be employed outside the setting of neoadjuvant trials. 

### 2.3. Profiling across Preclinical and Clinical Samples

In cases of rare tumor types or exceptional clinical responses, it can be useful to perform case-study analyses to generate hypotheses about the molecular drivers of the unusual features. These case studies can benefit from less biased profiling strategies in order to maximize the amount of information collected from the individual or small cohorts of samples. In addition to profiling human tissues, the GeoMx platform can be employed to characterize preclinical models [69,70]. The experimental design strategy can be adapted to study organoids or in vitro reconstructed tissue models using the same reagents that are used on human tissues. Protein and RNA reagents are available for characterizing tissue samples from murine model systems, including syngeneic tumor models, xenografts and genetically engineered models. In some model systems such as murine xenografts, human and mouse-specific profiling morphology reagents can be used to discriminate the origin of species-specific expression profiles. By using a common platform to characterize both preclinical and clinical samples, observations can be translated between systems, allowing for rapid hypothesis testing and confirmation, accelerating scientific discovery. 

## 3. Applications of GeoMx DSP in Breast Cancer

The GeoMx platform has been used to address questions specific to breast cancer, providing insight into pharmacodynamic effects, preclinical models, treatment response, treatment resistance and prognosis. As examples in real-world application of this technology, selected published studies in breast cancer research are briefly summarized in this section.

### 3.1. Characteristics and Spatially-Defined Immune (Micro)Landscapes of Early-Stage PD-L1-Positive Triple-Negative Breast Cancer (Carter JM et al., Clinical Cancer Research 2021) 

Carter et al. applied GeoMx to a cohort of untreated PD-L1+ and PD-L1− TNBC to profile the tumor-associated immune response and identify candidate therapeutic targets [71]. A total of 184 tumor cores were assembled into a TMA and characterized with a panel of 58 antibodies. A single ROI of 600 µm was placed within each tumor core, segmented into PanCK+ intraepithelial tumor compartment from the PanCK- stroma and each segment profiled independently. 

They observed that PD-L1+ tumor and stroma had greater expression of immune proteins, including IDO1, HLA-DR, CD40 and CD163, as well as stroma specific alterations in CLTA-4, STING and fibronectin. The profiling was validated orthogonally by comparing the PD-L1 counts obtained from GeoMx DSP to PD-L1 IHC with the FDA approved SP142 and 22C3 companion assays and high concordance (>95%) was observed between the two platforms. PD-L1 was observed to be expressed at low levels and was correlated most strongly with myeloid-associated proteins, including CD14, CD40, CD68 and CD163. Additionally, there was higher expression of T cell-associated proteins including CD3, CD4 and CD8 in the PD-L1+ segments, as potential therapeutic targets, including STING, VISTA and TIM-3, were expressed at higher levels in the PD-L1+ segments, whereas CTLA-4 was decreased in the PD-L1+ stromal segments. In summary, this paper identified different immune responses between PD-L1+ and PD-L1− TNBC that may be exploited in the future to develop more effective immunotherapies and associated biomarkers of response.

### 3.2. Spatial Proteomic Characterization of HER2-Positive Breast Tumors through Neoadjuvant Therapy Predicts Response (McNamara KL et al., Nature Cancer 2021) 

The McNamara et al., 2021 study is a good example of *profiling across time* to discover biomarkers that change with treatment and predict patient outcomes [35]. The study used archival FFPE tissue from the HER2+ neoadjuvant breast cancer cohort TRIO-US-B07 that included three timepoints, pre-treatment baseline biopsies, run-in biopsy after short-term HER2-targeted therapy and post-treatment surgical tissue after combination chemotherapy with HER2-targeted therapy. Using a 40-plex GeoMx DSP protein assay, the study spatially resolved protein expression in the tumor and microenvironment compartments. Their results included the identification of the CD45 immune biomarker within the tumor-enriched regions in the short-term HER2-targeted biopsy timepoint, which was associated with the later patient outcome of pathological complete response.

### 3.3. Spatially-Resolved Quantification of Proteins in Triple Negative Breast Cancers Reveals Differences in the Immune Microenvironment Associated with Prognosis (Stewart RL et al., Scientific Reports 2020) 

Building on the observations that immune responses in breast tumor are clinically important, Stewart et al. used GeoMx DSP to correlate the presence and abundance of immune proteins with prognosis of TNBC [46]. Ten untreated TNBC tumors were characterized by GeoMx using PanCK, CD3 and a nuclear stain as morphology markers. Three stromal and three epithelial regions were selected in each tumor using a geometric AOI, from which expression of 39 protein targets was quantified. Selected results were validated orthogonally with single-plex IHC. 

They observed a subset of samples that expressed HLA-DR in the tumor compartment and those same samples also had high expression of CD4 and ICOS in the stromal compartment. Furthermore, tumors were stratified by relapse status of the patients and differential expression analysis was performed to identify targets associated with relapse or disease-free survival (DFS) after 7.4 years. ICOS, CD45, CD11c, CD3 and CD8A in both the tumor and stromal ROIs were all associated with DFS, whereas CD45RO, CD4, PD-1 and CD20 in the stroma were associated with DFS and HLA-DR, IDO1 and B2M in the epithelial ROIs were also associated with DFS. These results demonstrate that even from a small number of samples, GeoMx profiling can be used to identify potential prognostic biomarkers that may have clinical relevance. 

### 3.4. Multiplexed Digital Spatial Profiling of Invasive Breast Tumors from Black and White Women (Omilian AR et al., Molecular Oncology 2021) 

In this study, GeoMx DSP was used to profile tumor- and immune-related proteins in FFPE TMA tissue sections of invasive breast tumors from black and white women [45]. Dr. Omilian’s group previously profiled gene expression in tumor samples from a subset of cases in this multi-site study using the nCounter PanCancer Immune Panel. To profile these tissue samples deeper, GeoMx DSP was performed to simultaneously assess 52 analytes within spatially-resolved tumor and stromal compartments that were defined by PanCK expression. For this experiment, FFPE TMA sections of invasive breast cancer cases were incubated with a cocktail of 58 barcoded antibodies.

GeoMx DSP analysis showed robust protein signal in 33 analytes and the results were highly reproducible. For a subset of markers, correlative analyses showed from moderate to very strong associations between DSP analytes and traditional IHC scores. Similarly, DSP analytes and gene expression scores were concordant for 21 of 25 markers. Among the 25 immune markers tested, 14 markers had a significant inverse association with expression of estrogen receptor. Moreover, the scores of the B7-H3 protein were significantly lower in breast cancers from black women than in those from white women. GeoMx DSP markers that were associated with survival included CD8, CD25, CD56, CD127, EpCAM, ER, Ki-67 and STING. Thus, DSP was used as an efficient tool for simultaneous screening of tumor- and immune-related markers and the results were concordant with established immune profiling assays.

## 4. Considerations for Robust Experimental Designs 

The quality of the data produced by spatial profiling is as good as the experimental design that is employed in the data collection. Therefore, it is imperative to carefully consider all aspects of how the data is generated, collected and analyzed prior to initiating the study. In this section, we have outlined the sample, ROI placement and workflow considerations for GeoMx experimental design that impact data quality. 

### 4.1. Sample Considerations 

#### 4.1.1. Sample Preparation and Tissue Size 

For GeoMx-based spatial profiling of RNA or protein, both FFPE and frozen tissues can be used [44]. Most frequently, spatial profiling is performed on standard ~5 µm sections cut from FFPE tissues. This is a major advantage, as the tissue requirement for GeoMx DSP is minimal and archival clinical FFPE-based specimens. 

For GeoMx profiling, tissue samples should be mounted onto standard, positively charged glass slides, such as Superfrost^®^ Plus Micro Slide or comparable [72]. Of note, when profiling RNA in FFPE tissues, sections should be freshly cut off the block (i.e., less than 30 days) prior to the study in order to reduce oxidative damage. Sections stored for longer periods of time would likely suffer from reduced signal-to-noise but may yield sufficient information, depending on storage conditions. If the tissue sample has low cellularity (i.e., non-neoplastic breast tissue) and is prone to detaching from the slide during routine processing, tissue sample adherence may be improved by using Apex BOND Adhesive slides or Leica BOND Plus slides. Importantly, in all cases, the region to be captured for GeoMx DSP must be located within a specific area of the glass slide (Figure 2) [72]. 

In tumor studies, CNBs and resected tissue samples are the most frequently available samples for spatial profiling. CNBs are usually 1–2 mm wide by tens of millimeters long with a limited amount of tissue that contains variable mixtures of tumor and adjacent normal cells. Spatial profiling methods such as GeoMx are ideal to maximize the amount of information that can be generated from such a single small tissue section. CNBs are particularly well suited for analysis of phospho-proteins as the short ischemic period preserves post-translational protein modification. However, interpreting data from CNBs can be challenging because the sample may not be representative of the TME in the entire tumor (i.e., “sampling errors”). 

Tissues samples from surgically resected tumors, made available for profiling when in excess to pathological assessment, better represent the architecture of the TME and may also contain adjacent non-malignant tissue. In any case, a single section from the tumor captures only a fraction of the biology in the whole tissue, so results should always be interpreted conservatively and/or significant findings validated on a separate tissue section. 

#### 4.1.2. Sample Tissue Features

In breast cancer, there are multiple classification systems to describe tumor types and guide the choice of treatment regimen, including histology, grade and receptor expression (estrogen receptor (ER), progesterone receptor (PR), HER2), and also to help for prognostic evaluations such as intrinsic molecular subtypes (Luminal A, Luminal B, HER2 enriched, basal) [73]. Histopathological evaluation classifies tumors into a number of subtypes, such as invasive ductal carcinoma (IDC), invasive lobular carcinoma (ILC), ductal carcinoma in situ (DCIS), lobular carcinoma in situ (LCIS) and others [74,75]. These subtypes should be taken into consideration when selecting tissue samples and ROIs in spatial profiling experiments. For instance, a tumor sample from an individual patient often contains mixed histopathological types, e.g., IDC and DCIS, in addition to normal breast tissue (Figure 3). 

### 4.2. ROI Placement and Compartmentalization/Segmentation Strategies

When considering the placement of ROIs for GeoMx studies, it is necessary to evaluate the underlying tissue features that may be reflected in the data. A great deal of work had been undertaken by other groups to define each of the tissue features and to map them to the tissue architecture, including work by the NIH Common Fund Human Biomolecular Atlas Program (HuBMAP)’s common coordinate framework and the Human Cell Atlas (HCA) community [76,77]. Since tumor architecture has traditionally been defined through hematoxylin-and-eosin (H&E)-stained sections, it is helpful to have H&E-stained sections of the tissue to guide ROI placement prior to a GeoMx run. 

Accurate placement of ROIs and compartmentalization of ROIs and AOIs can greatly influence the quality of data and the ability to answer the given research question. Therefore, a skilled breast pathologist is an essential part of the research team for GeoMX DSP studies. For analyses, ROI selection and compartmentalization strategies should be consistent within experiments and across experiments (i.e., standardization) to allow cross-study data comparisons. 

#### 4.2.1. Selecting Visualization Markers 

For each experiment, up to three fluorescent-tagged antibodies (i.e., morphology markers) along with nuclear dye can be used to (1) facilitate selection of ROIs and (2) guide segmentation. For example, the PanCK antibody highlights epithelial cells and is frequently combined with pathologic assessment for selection of cancer-based ROIs and for distinction from benign tissues (Figure 3). Depending on the experimental strategy, additional morphological markers can be used to guide ROI selection and segmentation (e.g., immune-based markers such as CD3, CD8, or CD45, or breast biomarkers of interest such an ER, PR, or Ki67) to target specific tumor regions. 

When profiling immune cell-infiltrated regions, the density of CD45 and CD3 can distinguish areas of high and low immune infiltration within the tumor and often identifies areas with tertiary lymphoid structures [78]. In addition, morphology markers specific for other immune cell types, including B-cells, natural killer cells, M1 and M2 macrophages and dendritic cells, can be used to target tumor or the TME enriched in these immune cell populations [42,79]. For cross-study comparisons, the selection and use of fluorescent morphology markers should be as consistent as possible to properly align ROI categories across cohorts. 

#### 4.2.2. ROI Placement and Tissue Considerations

There are a number of key macrostructures and compartments within the breast TME and surrounding tissue that can be profiled to better understand tumor heterogeneity. Most frequently, tumor regions are chosen for profiling based on pathological characteristics or tumor subtypes with the scientific question in mind, such as neoplastic regions, DCIS, metastatic lesions, invasive margin (IM), or tumor center (CT). When profiling immune cell infiltration, selecting ROIs in the IM versus the CT can identify mechanisms related to epithelial–mesenchymal transition, invasion and metastasis. Detecting T cell populations allows the measurement of TILs and the resulting “immune hot”, “immune excluded”, or “immune desert” phenotype [80]. 

To identify quality samples and ROI locations, it is helpful to guide the selection of pathological regions using archival H&E images or H&E collected on sequential sections. For profiling heterogeneity associated with variability in HER2, ER, or PR expression, standard pathology IHC images run on serial or nearby sections can guide selecting ROIs that have heterogeneous staining of additional markers within a single TME. 

If possible, it is important to select ROIs of similar type (e.g., invasive margin, immune hot, immune cold, etc.) across all tissues to compare across samples. In particular, when inspecting patient-matched samples (e.g., primary versus metastasis, pre-, on- and post-treatment), selecting similar ROI placement/meta characteristics (e.g., tumor-immune hot, tumor-immune cold, tumor-high proliferation) in both samples allows researchers to compare differences across the samples. There are situations when it is not possible to select replicate ROIs of a single annotation. For example, when examining pre- and post-treatment samples from patients who experienced pathological complete responses (pCRs) or near pCR, minimal or no residual cancer is available in the post-treatment samples. In these cases, it is recommended to capture as much tumor as possible. If no tumor is present, these ROIs cannot be directly compared to pre-treatment tumor samples. In addition, when working with a TMA, it is important to select ROIs or a segmentation strategy to maximize the cellularity of tumor (or cell type of interest) and maintain a consistent ratio of tumor and stroma as much as possible. If the area of the core allows it, multiple regions from a single core can be selected to capture more of the tumor and/or microenvironment.

Whenever possible, tissue areas or defects that may cause non-specific antibody or in situ RNA probe-binding should be avoided, such as tissue folds, adipose tissue and non-viable or necrotic tumor regions. Similarly, hemorrhagic regions or regions with concentrated red blood cells tend to emit autofluorescence in multiple fluorescence detection channels and may confound fluorescence-based masking during ROI selection, but it has not been shown to affect GeoMx RNA analysis.

#### 4.2.3. ROI Size and Quantity

The most important factor for selecting ROI size is the number of cells per AOI. There is an inherit correlation between the size of an AOI and the number of transcripts or proteins available for measurement. To capture the full dynamic range of expression and maximize cross-cohort analysis, it is recommended to include at least 50 cells in an AOI for protein and at least 100 cells for in situ RNA, regardless of segmentation strategy. Including more cells in an AOI allows more targets to be detected above the background and the limit of quantitation, though lower numbers of cells may still result in the detection of high expressors [42]. A large AOI area can be used as the background for RNA normalization (see “Section 5.1.1. Data Normalization”). Above all, it is ideal to keep consistent ROI size and cell numbers across a study. 

To capture tumor heterogeneity, a minimum of 3–5 replicates per given annotation within ROI (e.g., IM, CT, tumor, immune hot, immune cold) should be selected to enable robust statistical analysis [42]. If profiling heterogeneous structures (e.g., including vessels and stromal tracks), more ROIs may be needed to fully assess the diversity. Furthermore, morphologically distinct regions within tumors should be sampled as separate ROI classes in case they molecularly diverge. Finally, it is helpful to annotate ROIs during selection with standard and descriptive language to describe ROI morphology, which is critical for ensuring robust downstream analysis.

#### 4.2.4. Area of Illumination (AOI) Segmentation Strategies

After selecting ROIs, tissue and structures within ROIs are profiled by several AOI segmentation strategies: geometric, tumor and stroma segmentation, cell-type specific, contour and complex segmentation profiling. Table 1 and Figure 4 describe the different AOI segmentation strategies and parameters around ROI selection and feasibility of metadata integration. 

**Geometric profiling:** Geometric AOIs profile the entire area within an ROI and uses standardized geometric shapes or custom polygons across distinct tissue regions. This strategy is useful for characterizing different classes of megastructures to measure tissue heterogeneity at different locations within and near a tumor. If the researcher is interested in only tumor-related components (e.g., ER expression, HER2 status, Ki67 expression) or breast tumor types that have little or no immune infiltration, geometric ROIs in the tumor areas would be a simple, optimal selection. In addition, large geometric ROIs are useful for profiling the areas with fewer viable cells. Tissue is profiled in an unbiased manner by placing geometric ROIs at regularly defined intervals across the sample (i.e., gridded profiling).

**Segment profiling:** Segmentation of cellular compartments within an ROI is another simple method included in the GeoMx software. This segmentation technique allows the quantification of markers in distinct biological compartments within an AOI identified using morphology markers. In breast cancer, a typical application for segment profiling is analysis of tumor versus stroma where malignant cells and the surrounding stromal cells are segmented based on histopathologically identified PanCK status [41]. Separate profiling of the tumor and stromal AOIs is most useful when interrogating breast cancer samples with immune cells in the stroma. Furthermore, the GeoMx system offers an extensive flexibility in AOI selection and segmentation through multiple adjustable parameters: (1) segment definition, (2) erosion, (3) N-dilation, (4) hole size, (5) particle size, (6) collection order and (7) threshold (Table 2). 

To summarize segment profiling strategies, a researcher can first tailor the fluorescent thresholds that define an AOI mask. The thresholds can be adjusted on an ROI basis and enable histopathologically informed masking of the desired cell population. After adjusting thresholds for each channel by ROI, segmentation should be followed by global parameters, such as hole-filling of nuclei when IF staining is primarily membranous, particle exclusion to remove artifactual background staining, or partial cell capture during slide prep and/or increase pick-up of punctate cell types and erosion.

**Cell type-specific profiling:** Another common experimental design is to leverage the in situ profiling of a specific type of cells throughout a tissue. Cell type-specific profiling is a variation of the segmentation strategy that characterizes expression changes within the specific cell population (not necessarily contiguous) (Figure 4). To understand which genes are expressed by a specific type of cells, integration with additional data, such as reference AOIs capturing the cell type-specific microenvironment or single-cell RNA sequencing data, can help qualify the target expression within these AOIs [81]. As this profiling method is sensitive to protocol and reagent variations, coordination between sites is critical for overcoming the challenge of meta-analysis. A more readily implemented alternative is to rely on a broader profiling strategy and leverage cell deconvolution approaches to gain insight into cell-specific activity while studying the broad TME simultaneously.

**Contour profiling:** Contour profiling measures gradients and distances from megastructures to examine how proximity affects biological response and the local microenvironment around a central structure [41]. In this strategy, a megastructure is identified within the ROI, then AOIs at fixed distances are serially illuminated and the released oligos are individually collected. For example, contour profiling can be used to evaluate expression gradients at different distances from the tumor IM and/or CT (Figure 4). The contouring widths can be varied between small distances (10–20 µm) and larger contours (50–100 µm). To profile how signaling spreads as a function of distance from the boundary, it is recommended to capture at least three AOIs on either side of the boundary. Certain advanced workflows in image analysis such as this are accessible by using ImageJ or an additional digital pathology software.

**Complex profiling:** Complex profiling represents combinations of segments, cell type-specific and contour profiling to enable characterization of multiple and disparate AOIs within an ROI. An example of this would be to use segmentation profiling to characterize all of the tumor cells in an ROI, then cell-type specific profiling to characterize the T cells within the tumor and stroma (Figure 4). 

### 4.3. Workflow Considerations

During experimental design, the project goals and scientific questions should be reviewed to ensure the workflow supports answering those questions. If it is a pilot or exploratory study to determine if there are differences between cohorts, a smaller experiment may be suitable. By contrast, an in-depth biomarker discovery project or deep characterization of tissue biology may require a large-scale experiment. 

There are a number of factors that influence the GeoMx workflow (see Figure 1). For slide preparation, the protocol is determined by the type of analyte assay (RNA or protein), manual or automated slide preparation and morphology markers. For the GeoMx instrument run, the run time is influenced by the number of slides, ROI selection strategy and total number of AOIs collected. Finally, the readout platform workflow (nCounter or NGS) is established by the number of collection plates and, for NGS, the sequencing depth needed for the GeoMx panel (Whole Transcriptome Atlas (WTA), Cancer Transcriptome Atlas (CTA), or RNA Panels). Each parameter determines the protocol and time of execution.

The GeoMx instrument run is an important part of the workflow to consider for execution time (see 2–5 in Figure 1). GeoMx houses an instrument tray that holds up to four slides per run and collects samples in a 96-well collection plate, with each well corresponding to sample aspirate (photo-cleaved oligo barcodes) from a given AOI. Each slide takes approximately 20–30 min to image, depending on the tissue size. Experiments that have more than four slides on one run require additional time for reloading the slide tray. 

After image acquisition, the researcher selects ROI on the tissue and submits the AOIs for automated collection. Each cycle of AOI collection takes about 2 min. If more than one microtiter collection plate is needed during a run, then each collection plate switch adds between about half an hour and one hour to the workflow time. Taken together, these time considerations help define the throughput of one GeoMx instrument run and guide decisions of the number of samples, slides and total AOIs collected. 

For the NGS readout, consideration of the sequencing strategy helps ensure cost-effective sequencing and optimal read depth. Sequencing depth for an experiment is estimated by multiplying the total area of all the AOIs (µm^2^) with the sequencing depth factor for the GeoMx panel. For this estimation, a forecast of approximate AOI size, as well as how many AOIs collected, is needed. Alternatively, targeting specific NGS instruments and flow-cell capacities first, then back calculating how many AOIs of a particular area can guide experimental design. For example, a NextSeq500/550 high output v2.5 flow cell with a capacity of 400 million clusters is estimated to fit about 44 AOIs of 300 µm^2^ size of the WTA panel, considering the total area (4,050,000 µm^2^) and sequencing depth factor of 100 clusters per µm^2^. In addition, up to 768 AOIs may be uniquely indexed with the GeoMx NGS readout assay and these AOIs can be pooled and sequenced on a single flow cell.

When increasing sample throughput, there are key strategies to consider; these depend on the analyte of interest, the tissue-slide scan area and instrument setup. Slides incubated with GeoMx protein-detection antibodies can be batch-stained, mounted with Fluoromount-G^®^ and stored in the dark at 4 °C for up to 3 months without loss in signal. This scalable approach minimizes the number of days dedicated to slide preparation across a large experiment and enables the collection of data from the GeoMx across several consecutive days. The mounting method, however, does not apply to slides stained with RNA panels due to diminishing mRNA quality and signal over time. When considering the large scan area available to profile a slide, multiple tissue sections from different blocks are placed onto one microscope slide. This greatly increases tissue sample throughput, but care should be taken when arranging variable sized tissue on a single slide to fit within the gasket. In this scenario, patient and slide effects should be incorporated separately into any downstream analysis. Formal implementation of the multi-tissue strategy is the use of a TMA, to profile 100+ tissue cores from an individual slide. 

For large studies, it may be necessary to profile the samples over a period of time, or in multiple batches. In these cases, care should be taken with the experimental design to minimize any batch effects that may arise over the course of data collection. Samples should be run with a single lot of reagents if possible. If not, a subset of samples should be run on both reagent lots to permit batch correction during data analysis. Additionally, the samples should be run in a way that minimizes any technical variability between groups; for example, by running control samples side-by-side with case samples or randomizing the run order for two sample cohorts. 

## 5. Tools for Data Analysis and Image Sharing

GeoMx studies combine multiplexed IF with quantification of proteomic and transcriptomic states into a single analysis. As the platform has advanced since its launch, new capabilities have been developed to better integrate GeoMx into pipelines that fit into the native ecosystems for these data types. The GeoMx Data Analysis suite provides a range of native software tools for analysis, ranging from exploratory tools such as principal component analysis (PCA) and clustering to more advanced data modeling, including linear mixed effect models and pathway analysis through gene set enrichment analysis (GSEA) [82,83]. This suite allows local and remote access to datasets collected over the lifecycle of a study. Moreover, as studies designed to integrate across multiple machines or sites as laboratory cores are conducted, new data exchange formats have been built to support the integration of studies. To allow integration across machines, GeoMx supports the direct transfer of integrated count and IF image data. 

Additional support for off-instrument analysis has also been expanded and provides new flexibility for data analysis and cross-site collaboration. For IF images, GeoMx supports analysis of full-scan images both within the software and with third-party software. After slides are scanned, images may be exported in the OME-TIFF format which can be leveraged by digital pathology suites such as quPath, Visiopharm, Halo and others. When exported, these images contain all the scan information and metadata related to scanned image and IF channels, as well as the location and segmentation of ROIs. In addition, the molecular count data can be analyzed using methods that have been developed to interrogate bulk profiling expression data. NanoString has recently published a data analysis package on Bioconductor, GeoMxTools [84], which may be used to analyze data using statistical pipelines, in addition to the fully-featured analysis suite on the machine. 

Data sharing is planned for multiple data types throughout the GBCC lifespan. During the initial stages, de-identified metadata from breast cancer cohorts or samples should be available to the consortium to facilitate the design of collaborative studies. As the datasets mature, access to secondary analysis and summary results form a backbone of data sharing, from which researchers can learn about ongoing experiments. Throughout the experimental cycle of the studies, holistic data packages conforming to best practices in the field shall be organized and submitted to appropriate open databases. Using standard data formats, such as OME-TIFFs (TIFF image files that contains an OME (Open Microscopy Environment)-XML metadata block) and MIAME (Minimum Information About a Microarray Experiment)/MINSEQE (Minimum Information About a Next-generation Sequencing Experiment) compliant data, is recommended for all data types generated to allow the furthest reach across communities. 

### 5.1. Analysis Guidelines for GeoMx Studies

#### 5.1.1. Data Normalization

Analysis of GeoMx data requires the application of both quality control measures and normalization strategies that depend on the assay type. Samples with poor target detection, low signal-to-noise ratios, or low nuclei counts may be flagged and excluded from analysis. In all experiments, consideration must be given for the appropriate normalization method to prevent misinterpretation of the data. 

For small protein or RNA panels (*n* < 300 targets), housekeeping (HK) normalization or negative control normalization is the preferred method to normalize data. For HK normalization, protein panels include 2–3 endogenous housekeeping proteins (S6, Histone H3, GAPDH). These proteins measure on-target antibody signal and S6 and Histone H3 are typically selected for breast cancer-related GeoMx studies. While this decision should be made within each study by analyzing the pairwise correlations of the three HK proteins, GADPH has been less concordant than the other two proteins in studies of breast cancer samples. An alternative strategy is background normalization performed with the immunoglobulin G (IgG) negative controls. Background normalization requires the IgG counts be appropriately detected and correlated with each other. AOIs with low IgGs counts (<20 counts) are not as suitable for background normalization, as small deviations in IgG values profoundly affect the endogenous protein counts. Other methods, such as quantile normalization or normalization to the geometric mean of all non-control probes, may be appropriate if HK or IgG normalization is not suitable [85]. 

When using large RNA-profiling panels, such as the CTA or WTA panels, expression data can be more formally analyzed than on smaller panels, or than protein data, where each probe may have different dynamic ranges of expression due to distinct binding characteristics. Normalization strategies that have been used in RNA-Seq, e.g., upper-quartile or trimmed mean of M values (TMM) normalization [86], may be sufficient in some cases but do not account for the dynamic range of counts specific to the GeoMx platform. Modeling approaches in single-cell and bulk RNA-Seq have shown that such an approach can also be used to directly account for count distributions [87,88]. Similar approaches are appropriate for GeoMx DSP to account for the data distribution and count dispersion characteristics from sequencing. Such methods shall be natively incorporated into future releases of the GeoMx data analysis software and will be released on Bioconductor as analysis packages for GeoMx DSP data. Other normalizations strategies that rely on surrogates for background, such as nuclei count or area, are not recommended, as the inclusion of negative control probes provides direct ways to estimate background in a segment-specific manner.

#### 5.1.2. Analytical Considerations for Running Single- and Multi-Site Studies

To leverage the data generated during GeoMx studies to understand differential gene expression or protein expression between biological compartments and across patients, there are a few best practices of analytical considerations. For large-scale translational analysis that examines cohort-level patient heterogeneity, in addition to the spatial organization of the TME, it is most appropriate to use approaches that account for both intra- and inter-patient variation. This type of studies frequently sample multiple times from the same class or type of ROI or segment for each patient. Such methods include a class of analyses called mixed effect models, which are natively implemented in the data analysis suite, as well as easily integrated into data analysis pipelines with the GeoMxTools package. In addition, these methods can be used to account for batch-specific effects in order to control for technical artifacts in the data, site-to-site variability, different lots of reagents, or an instrument being serviced during the course of a study.

For each class of ROIs, each patient or slide has some level of variation in the counts which can be estimated by mixed effect models. Rather than using the mean or median expression level within a class and patient grouping, this variance can inform the significance of an observed association. Approaches such as basic *t*-tests or ANOVAs should be limited to studies that do not have replication within a single slide. Even for studies using TMAs that range across different slides or sites, a mixed effect model should be used to account for slide-to-slide variability. Meta-analytical approaches of these models are suited for the integration of multi-site studies to understand a variation in samples across different institutions. 

Survival analyses should leverage mixed effect models and account for the multiplicity of testing within and across samples. Common statistical analytic approaches (e.g., Kaplan–Meier survival curves) may not account for ROI multiplicity and should be avoided, unless summary statistics (e.g., median) are used to represent all ROIs for a patient prior to survival analysis. As with other data types, post-hoc data qualification, such as confirming target expression levels above background, as well as orthogonal techniques for validation, may be recommended to provide further evidence to the significance of relevant targets. Data from targets with low expression, at or near the background level, within a comparison class, should be interpreted with caution. In this case, quality control using an MA plot, which plots mean expression against fold change for genomic data [89,90], or graphing the signal-to-noise ratio of a target may help qualify significant findings. Some analyte-specific or design-specific considerations are outlined in the following section.

#### 5.1.3. Interpreting a Protein’s Counts from an AOI

When interpreting the signal for a given protein target, it is important to recognize that each antibody is characterized by its own dynamic range of on-target/off-target binding. Although isotype control antibodies are included with every GeoMx run to measure non-specific binding, antibodies can exhibit an intrinsic background that is innately higher or lower than these negative controls. During development, antibodies must be validated against control samples with known expression patterns of the antigen. This is routinely performed for commercial GeoMx antibodies but must be independently validated by the user for any custom-labeled antibodies. Furthermore, since every antibody is unique, their expression patterns should be assessed with an understanding of relative expression within and across the cell populations under study and the relative expression of different targets cannot be compared directly due to inherent differences in the binding affinity of the antibodies. 

#### 5.1.4. Comparing Pre- Versus Post-Treatment Data

Interrogating both pre- and post-treatment tumor samples is a key experimental goal in translational research of breast cancer. However, pre- and post-treatment cohorts may need to be analyzed independently due to sample preparation discrepancies, sampling location misalignment, ROI selection limitations, or other analytical considerations which influence GeoMx DSP experimental design. For example, while pre-treatment samples may have sufficient tumor and stromal immune cellularity to merit a tumor/stroma segmentation strategy with GeoMx, treated tumors may lack sufficient residual tumor or have depleted immune reservoirs. 

Even when a consistent ROI selection strategy is possible, pre- and post-treatment sampling methods may differ and this sampling difference introduces potential bias in differential expression results. Simple analyses such as dimension reduction with PCA, t-distributed stochastic neighbor embedding (tSNE), or uniform manifold approximation and projection (UMAP) analysis can help identify whether the timepoint-specific expression is confounded with the type of tissue collected or the amount of residual disease. If timepoints and these characteristics are confounded, it may be necessary to segregate some of the tissues during analysis, focusing on pre- and post-treatment cohorts independently. This allows for focused interrogation of predictive or prognostic biomarker identification on the pre-treatment samples, while exploring PD modulation in post-treatment samples, even if the target tumor tissue is no longer present. As such, when exploring therapeutic intervention across timepoints, it is recommended to identify PD biomarker surrogates for drug activity as key hypotheses to test and understand tumor clearance and response to the therapeutic intervention. If the timepoint does not appear confounded by sample collection type or residual disease, then fully integrated analysis can be pursued to better characterize samples and explore expression dynamics in matched tissue samples.

#### 5.1.5. Reference Sample Inclusion in Ongoing Studies

To address possible technical artifacts associated with different batches of samples run at multiple sites or across reagent lots, it is critical to include standard reference samples within and across studies. Such analytical components, such as controlled cell-pellet arrays or reference tissues, aim to provide clear mechanisms for normalizing for technical artifacts. The use of these reference samples in combination with advanced modeling of sample-specific biases can potentially normalize any analyte or sample-handling runs, but it needs to be qualified across a large array of platform runs. This is an area of active investigation by GBCC members. As reagent standards are established and qualified as measures of assay reproducibility and dynamic range, they may serve as a way of tracking long-term trends within and across study sites.

### 5.2. Integration with Digital Pathology Toolkits

To best leverage and integrate the digital images of slides collected with GeoMx and the corresponding quantified GeoMx data, a common platform and toolkit is required for analysis. Open digital pathology toolkits, such as QuPath [91], can allow multiple sites to work using the same platform and automate and align image analysis methods for integration with the quantitative protein or RNA data. These platforms allow for quantification of the fluorescent signals within the sampled regions to determine cellular composition, cell counts and cell densities, which may be paired to the GeoMx-derived quantitative protein or RNA data. Additionally, co-registration of H&E section with the corresponding GeoMx digital image of the IF morphology markers facilitates either a priori or a posteriori analysis via artificial intelligence (AI) to guide ROI selection for profiling or qualify them after the collection has been completed.

In cross-site imaging studies, challenges for reproducible analysis stem from staining differences in samples and thresholding parameters used to tune cell segmentation methods (Table 2). Pairing H&E images with IF images may help qualify method performance and standard data samples run on instruments and shared between sites are the simplest path to mitigating site-specific image shifts. Ideally, these would incorporate more homogenous cell-based reagents, such as cell pellet arrays, to tune algorithms for use outside of the GeoMx infrastructure. As the GBCC members work together towards complex problems, data analysis methods should rely on rigorous tracking of covariates which might influence analysis. This includes site-specific information, as well as reagent-lot tracking and operators, for the best understanding of underlying differences between study sites.

## 6. Opportunities for Future Development

Although GeoMx is relatively well established for new technology, there are still opportunities for improvements that will enhance its functionality. One such opportunity is in the analytes that can be profiled on the platform. Although the whole transcriptome profiling capabilities for mouse and human samples provide complete coverage of the transcriptome for expressed genes, they currently do not address non-coding RNAs. However, some non-coding RNA species can be profiled using custom probes in parallel with the CTA/WTA assays. The protein menu could also benefit from additional reagents to detect more proteins, especially post-translationally modified proteins [92]. In addition, the development of detection capabilities for DNA sequences, such as somatic and germline mutations or copy number variations, would offer an intriguing dataset that could be integrated with the RNA and protein data. Some genomic alterations, such as copy number variation, can be inferred from high-plex expression panels such as the WTA.

To bring the most value from an expanding platform of molecular analytes, integration of data from multiple analytes, either from multi-omic studies or within the high-content assays, will allow for the development of complex signatures related to prognostic characteristics or intrinsic subtypes of breast cancer. While algorithms that estimate cell type abundance have begun being developed for GeoMx [81,93], the development of a more advanced algorithm that brings multi-gene signatures from bulk profiling to spatially resolved platforms is needed. 

The other aspect of spatial profiling is the role of digital pathology within the GeoMx platform. While the platform collects and provides access to high content imaging data in open formats compatible with digital pathology toolkits, it does not natively incorporate advanced AI methods into the workflow for collection. This is an area that could potentially improve both stability of ROI selection and reduce the workflow turn-around time. One potential path to harness this growing field of methodologies would be to integrate H&E image data or implement pseudo-H&E projection from IF stained slides. By mapping back to this standard for most digital pathology AI, the GeoMx platform could potentially use a wide variety of tools and platforms in clinical use or under development for use in breast cancer [91,94].

In parallel with improving the number of analytes that can be measured, improvements to the resolution of the platform could yield richer datasets. Near single-cell sensitivity for highly expressed proteins is achievable, but this is not universally true for all analytes [42,92]. Current recommendations of the number of cells to profile enable reasonably fine-grained characterization, although not to the level of single-cell profiling. However, with the proliferation of single-cell RNA-seq datasets in the literature and the development of cell deconvolution from GeoMx datasets, it is possible to generate analyses on GeoMx data equivalent to what is possible through single-cell profiling with the advantage of the spatial resolution. 

Sample compatibility and processing capacity are another area with opportunity for improvement. In terms of throughput, currently, the system can process up to eight samples a day if ROIs are preselected on an adjacent slide and the profiling strategy is not time-intensive. Furthermore, the current RNA assays require a freshly cut tissue section for profiling, as oxidation degrades the RNA. In contrast to the RNA assays, the GeoMx protein assays do work on properly fixed tissue sections that have been stored for longer periods of time, up to nine months at room temperature; thus, profiling opportunities exist for GeoMx, even if the sample has already been collected and mounted on a slide. 

Finally, the GeoMx platform, to date, has been used for research use only and has not been enabled for clinical applications. Nevertheless, spatial biology is an emerging and dynamic field where new developments and new discoveries will spur each other forward to advance our scientific understanding of breast cancer and the TME. 

## 7. Conclusions

GeoMx spatial profiling opens a new window into tumor biology and accelerates our understanding of the collaboration and competition between different cells in the breast TME across location and time. GeoMx-based spatial strategies can be used for clinical cancer research, as well as the mechanistic or functional studies in basic biology research. The potential identification of spatial molecular signatures associated with disease progression, response to interventions and patient outcome may enable the elucidation of tumor-specific biological pathways, including immune evasion, metastasis and drug resistance, leading to the development of effective novel therapeutics.

## Figures and Tables

**Figure 1 cancers-13-04456-f001:**
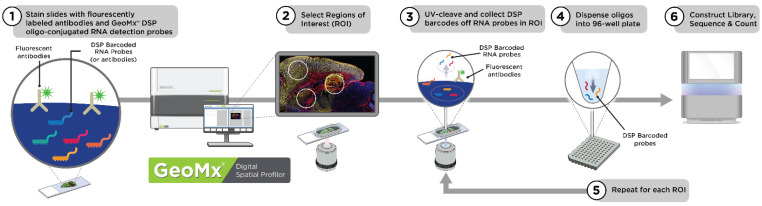
GeoMx workflow with NGS readout. The sequence of steps in the GeoMx workflow may be grouped into three phases: slide preparation (1), GeoMx instrument run (2–5) and readout (6). During slide preparation, a high-plex mixture of photocleavable oligo-linked probes (RNA shown) and morphology reagents are applied to the tissue section (1). Slides are loaded into the GeoMx instrument for a series of automated steps. After the researcher selects ROIs (2), the GeoMx instrument illuminates each area-of-illumination (AOI) with UV light to collect and deposit the photo-released oligos into a microtiter plate (3,4). Collection is repeated (5) to produce an AOI collection plate, where each well corresponds to a pool of photocleaved oligos from one AOI on the tissue. For NGS readout (6), each AOI (or well) is uniquely indexed during library preparation and can be pooled into one sequencing run.

**Figure 2 cancers-13-04456-f002:**
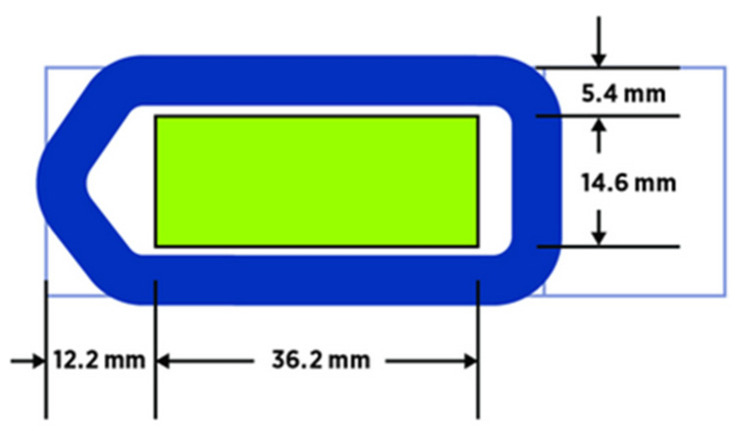
Slide dimensions and tissue placement for GeoMx DSP study. Tissue sections must be placed in the digital scan area (shown in green), measuring, at maximum, 36.2 mm long by 14.6 mm wide in the center of the slide. The tissue sections should not overlap the slide gasket (shown in blue) or the tip calibration area (the triangular region to the left of the green scan area). The frosted/label end of the slide is on the right.

**Figure 3 cancers-13-04456-f003:**
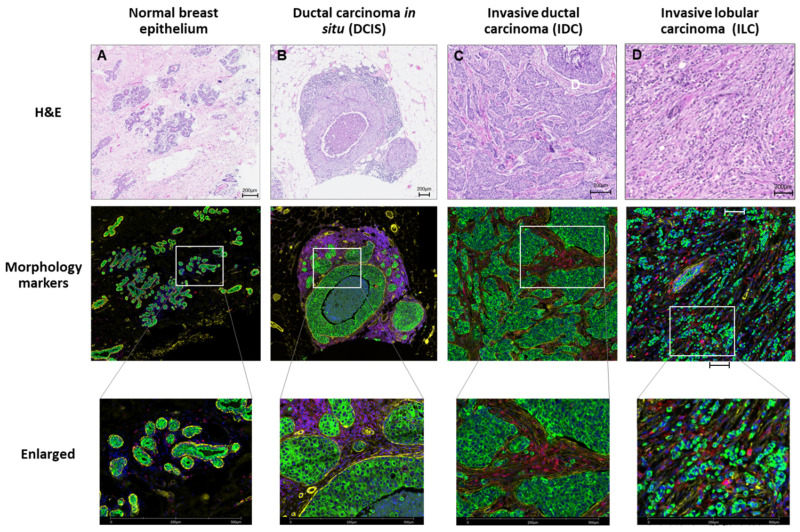
Representative images of non-invasive and invasive breast carcinoma based on histopathological subtypes. (**A**) Normal breast epithelium adjacent to invasive ductal carcinoma. (**B**) Ductal carcinoma in situ (DCIS) from concurrent invasive breast cancer. (**C**) Invasive ductal carcinoma (IDC). (**D**) Invasive lobular carcinomas (ILC). Hematoxylin and eosin (H&E) staining (upper panels). Morphology markers (middle and lower panels): PanCK (green), CD45 (red), smooth muscle actin (yellow) and DNA (blue). The boxed area is enlarged for better visualization. Scale bar: 200 μm.

**Figure 4 cancers-13-04456-f004:**
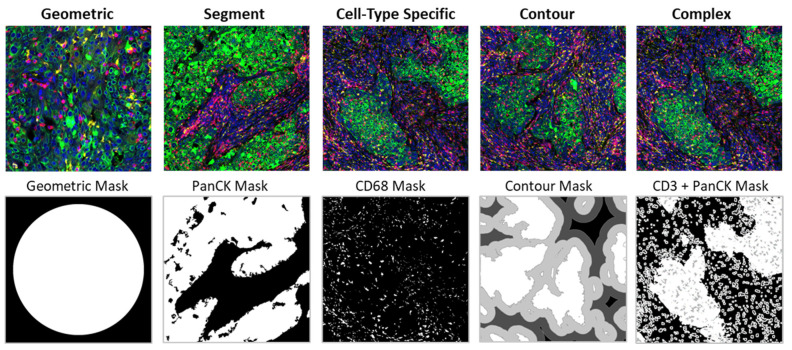
Representative images of AOI segmentation strategies in breast cancer tissues. The IF image was collected with the following morphology markers: PanCK (green), CD3 (red), CD68 (yellow) and DNA (blue). The mask image shows the captured AOI in white for geometric, segment, cell-type specific, contour and complex (PanCK+ tumor segment, CD3 cell-specific segment in the PanCK- stroma, CD3 cell-specific segment in the PanCK+ tumor and everything else) masking. For contour and complex masking, 4 separate masks are shown in grayscale.

**Table 1 cancers-13-04456-t001:** AOI segmentation strategies and parameters around ROI selection.

Profiling Type	Features	Variable AOI Area	GeoMx ROI Selection	Meta-Analysis Integration
**Geometric Profiling**	Assess tissue heterogeneity by profiling geometric shapes and polygons	Sometimes	Native software	Simple
**Segment Profiling**	Use morphology markers to identify and profile distinct biological compartments	Yes	Native software	Simple
**Cell-Type Specific Profiling**	Profile distinct cell populations with cell type-specific morphology markers	Yes	Native software	Difficult
**Contour Profiling**	Evaluate the proximity around a central structure using radiating ROIs	Sometimes	Custom	Moderate
**Complex Profiling**	Combine the above approaches to profile multiple cell types and/or complex regions of tissue	Yes	Native software and custom	Difficult

**Table 2 cancers-13-04456-t002:** Key adjustable parameters for AOI selection and segmentation.

Parameter	Features
**Segment Definition**	Segment definition determines the cellular composition of each AOI within an ROI based on the fluorescent morphology markers to profile distinct cell populations (e.g., CD3 for T cells, CD68 for macrophages) (Figure 4).
**Erosion**	Erosion enables the uniform removal of the UV-light mask boundaries that define an AOI. This parameter accounts for the subcellular spread in the focused UV light and is commonly set between 1 and 3 µm. For tumor AOIs that present with larger nuclei and more cell aggregation, erosion can be higher. For immune cell-specific AOIs that are more punctate in appearance, erosion is set lower to maximize signal and minimize removal of cells from the mask. To ensure the purity of the signal from the particular cell population, it can be helpful to erode the signal away from the cell membrane boundary by 2–5 μm to minimize contamination from surrounding cells.
**N-Dilation**	N-dilation uniformly expands the UV-light mask in an AOI, whose segment definition requires a positive signal for the nuclear marker (e.g., Syto13). This setting is only applicable for nuclear-tagged segments.
**Hole Size**	Hole size fills gaps in the AOI masks that are smaller than the value (in µm^2^) the researcher sets. This technique is particularly useful for segmenting tumor cells that have large nuclei that are negative for the tumor marker but are still intended to be part of the AOI of the tumor marker.
**Particle Size**	Any small segment areas (particles) less than this value (in µm^2^) are removed from the AOI mask. When segmenting immune cells, lowering the particle size value (to ~5 µm^2^) is important because a high value would subtract out immune cells from the mask.
**Collection Order**	Collection order determines the order in which UV light illuminates each AOI within an ROI. Since an AOI collection can be accompanied by minor signal loss near the AOI boundaries, it is valuable to collect AOIs from low to high abundance to maximize signal.
**Threshold**	Thresholds are adjustable per ROI and enable the researcher to tune the fluorescence-based masking that defines an AOI. This approach ensures specific illumination of target cell types or regions without bias or contamination. With the exception of thresholding, it is important to keep all other parameters above consistent across all the ROIs and slides in a given study.

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
