# Peer review of "Best Practices for Spatial Profiling for Breast Cancer Research with the GeoMx® Digital Spatial Profiler"

_cancers, 2021, doi:10.3390/cancers13174456_

Round 1

Reviewer 1 Report

The paper is well written but, to my opinion, the best practices you describe for breast cancer analysis using GeoMx DSP, are THE GENERAL BEST PRACTICES FOR WORKING WITH THE INSTRUMENT. They can and should be applied to any tumor.

However, I think that you should mention and cite the paper having PMID: 32458049 (McCart Reed et al: Digital spatial profiling application in breast cancer: a user's perspective. It was published in 2020 and presented first concrete experiences of GeoMx DSP users in breast cancer. The paper, although small, merits to be included in your review. You correctly cited the paper of Stewart RL et al (PMID: 32313087).

I also think it would be good to elaborate a little on GeoMx value in studying RARE breast tumors/cases, like exceptional responses/non-responses, where, for example, the "grid" scheme of regionalization could be very useful. GeoMx DSP is principally a research tool, so I think it should be exploited in studies of exceptional cases. Such studies teach us a lot, and there should even be a sort of financial support (grant calls etc) for studying such cases by GeoMx DSP instrument.

It would be also interesting (and useful) to elaborate a little on how to wisely do the segmentation in breast tumors with very little intratumoral stroma, for example, in many tumors of metaplastic histotype. OK, we can segment by cell type, or by localisation, but, well, metaplastic breast cancers are a challenge ... Maybe you can make a chapter "Challenges for GeoMx in breast cancer research" and elaborate a little on, for example, metaplastic cancers, phyllodes tumors, very "solid" cancers with almost no stroma etc ... Those are the situations where a wise GeoMx approach can reveal a lot, and other approaches can't ... This is just a comment/advice.

Author Response

Reviewer 1: Comments and Suggestions for Authors

The paper is well written but, to my opinion, the best practices you describe for breast cancer analysis using GeoMx DSP, are THE GENERAL BEST PRACTICES FOR WORKING WITH THE INSTRUMENT. They can and should be applied to any tumor.

Thank you for the suggestion. We absolutely concur with the reviewer that the platform has many uses in oncology research and beyond. Explicit reference to the application of GeoMx DSP in other cancer types is added to Lines 53-54, 69-71, 96-97, and 193-195 (with references).

However, I think that you should mention and cite the paper having PMID: 32458049 (McCart Reed et al: Digital spatial profiling application in breast cancer: a user's perspective. It was published in 2020 and presented first concrete experiences of GeoMx DSP users in breast cancer. The paper, although small, merits to be included in your review. You correctly cited the paper of Stewart RL et al (PMID: 32313087).

            McCart Reed et al. was cited with Stewart RL et al. in Line 193.

I also think it would be good to elaborate a little on GeoMx value in studying RARE breast tumors/cases, like exceptional responses/non-responses, where, for example, the "grid" scheme of regionalization could be very useful. GeoMx DSP is principally a research tool, so I think it should be exploited in studies of exceptional cases. Such studies teach us a lot, and there should even be a sort of financial support (grant calls etc) for studying such cases by GeoMx DSP instrument.

This is a very good point. We have now included discussion of the analysis of rare tumor types/cases in Lines 276-280, and the gridded profiling scheme is now added as a specific application of the Geometric Profiling strategy in Lines 569-570.

It would be also interesting (and useful) to elaborate a little on how to wisely do the segmentation in breast tumors with very little intratumoral stroma, for example, in many tumors of metaplastic histotype. OK, we can segment by cell type, or by localisation, but, well, metaplastic breast cancers are a challenge ... Maybe you can make a chapter "Challenges for GeoMx in breast cancer research" and elaborate a little on, for example, metaplastic cancers, phyllodes tumors, very "solid" cancers with almost no stroma etc ... Those are the situations where a wise GeoMx approach can reveal a lot, and other approaches can't ... This is just a comment/advice.

We appreciate your advice. These biological questions and challenges with the rare histological type of tumors are currently being explored by the consortium members, and we look forward to publications of their work to show how different profiling strategies can be informative.

Reviewer 2 Report

Bergholtz et al. have provided a well-written overview of the GeoMx Digital Spatial Profiler and applications for the technology in breast cancer research. The reviewer has a few suggestions to improve the article before publication. Major comments: 1. While this article is intended to be specific to breast cancer research, overall, it is somewhat generalizable to all the cancer research field, especially sections 3 and 4. To emphasize the utility of the platform for breast cancer research it is suggested that the authors include specific examples of how this technology has been used to address questions in breast cancer research. How has it provided insight into pharmacodynamic effects, preclinical models, treatment response, treatment resistance, or prognosis? These studies should be briefly described as real-world application examples. 2. Section 2.3: As much preclinical work in breast cancer research is conducted using murine models, a more thorough description and examples of how the technology can be used in these models and can separate the species-specific expression profiles (i.e. Human tumor vs. mouse stroma for xenografts) would provide a better overview of this specific application. Minor comments: 1. Figure 1 is much too small, please enlarge it for publication 2. In Figure 2 it is difficult to make out the blue stain in panels A and B and the yellow stain panels C and D. It would be helpful to include a magnified inset panel to be able to better visualize all the markers. 3. While AOI is described in the legend for Figure 1, please also define AOI in the main text. AOI and ROI seem highly similar and are used somewhat interchangeably. A better definition of what each term is and what are the differences between the two would help the reader better understand the terminology. 4. Line 419 should say (see “4.1.1. Data Normalization”). 5. Line 596, please define OME-TIFFs and MIAME/MINSEQE. 6. Line 665, define MA plot 7. Line 699, define “PD”

Author Response

Reviewer 2: Comments and Suggestions for Authors

Breast cancer, the leading cause of cancer diagnoses in women, is a heterogenous disease. Understanding the molecular diversity in breast cancer is critical for improving prediction of therapeutic response and prognostication. In this review, a group of breast cancer researchers (GBCC: GeoMx Breast Cancer Consortium) presents best practices for GeoMx profiling to promote the collection of high-quality data, optimization of data analysis, and integration of datasets to advance collaboration and meta-analyses. This review is well-structured, well-written, and easy to understand. It provides comprehensive and cutting-edge information and best practices regarding GeoMx-based spatial profiling of RNA or protein in breast cancer samples. It also addresses a subject that is of great interest in breast cancer community. Here are my some suggestions:

  1. In the section of Conclusions (Line #800), the authors stated that “GeoMx spatial profiling opens a new window into tumor biology and accelerates our understanding of the collaboration and competition between different cells in the breast TME across location and time.” Can the GeoMx spatial profiling be used for mechanistic or functional studies in basic breast cancer biology research? If yes, it is better to add this point.

Thank you for the excellent suggestion. The sentence regarding basic breast cancer biology research is now incorporated in Lines 930-931.

  1. The GeoMx spatial profiling should be also applicable for other tumor types such as lung or liver cancers. I would suggest adding a point regarding the application of GeoMx in other tumor types.

The sentence describing the application of GeoMx DSP in other cancer types is added to Lines 53-54, 69-71, 96-97, and 193-195 (with references).

Reviewer 3 Report

Breast cancer, the leading cause of cancer diagnoses in women, is a heterogenous disease. Understanding the molecular diversity in breast cancer is critical for improving prediction of therapeutic response and prognostication. In this review, a group of breast cancer researchers (GBCC: GeoMx Breast Cancer Consortium) presents best practices for GeoMx profiling to promote the collection of high-quality data, optimization of data analysis, and integration of datasets to advance collaboration and meta-analyses. This review is well-structured, well-written, and easy to understand. It provides comprehensive and cutting-edge information and best practices regarding GeoMx-based spatial profiling of RNA or protein in breast cancer samples. It also addresses a subject that is of great interest in breast cancer community. Here are my some suggestions:

  1. In the section of Conclusions (Line #800), the authors stated that “GeoMx spatial profiling opens a new window into tumor biology and accelerates our understanding of the collaboration and competition between different cells in the breast TME across location and time.” Can the GeoMx spatial profiling be used for mechanistic or functional studies in basic breast cancer biology research? If yes, it is better to add this point.

  1. The GeoMx spatial profiling should be also applicable for other tumor types such as lung or liver cancers. I would suggest adding a point regarding the application of GeoMx in other tumor types.

Author Response

Reviewer 3: Comments and Suggestions for Authors 

Bergholtz et al. have provided a well-written overview of the GeoMx Digital Spatial Profiler and applications for the technology in breast cancer research. The reviewer has a few suggestions to improve the article before publication.

Major comments:

  1. While this article is intended to be specific to breast cancer research, overall, it is somewhat generalizable to all the cancer research field, especially sections 3 and 4. To emphasize the utility of the platform for breast cancer research it is suggested that the authors include specific examples of how this technology has been used to address questions in breast cancer research. How has it provided insight into pharmacodynamic effects, preclinical models, treatment response, treatment resistance, or prognosis? These studies should be briefly described as real-world application examples.

This is an excellent point by the reviewer. As examples in real-world application of this technology, summaries of selected GeoMx breast cancer studies are incorporated in the newly created section “3. Applications of GeoMx in Breast Cancer” (Lines 292-384).

  1. Section 2.3: As much preclinical work in breast cancer research is conducted using murine models, a more thorough description and examples of how the technology can be used in these models and can separate the species-specific expression profiles (i.e. Human tumor vs. mouse stroma for xenografts) would provide a better overview of this specific application.

         The possible application of species-specific profiling is now mentioned in Lines 285-287.

Minor comments:

  1. Figure 1 is much too small, please enlarge it for publication

            Figure 1 is replaced with a high-resolution image with a larger font.

  1. In Figure 2 it is difficult to make out the blue stain in panels A and B and the yellow stain panels C and D. It would be helpful to include a magnified inset panel to be able to better visualize all the markers.

In the current Figure 3, the color balance of all the markers is adjusted, and a higher magnification image panel is added to each subtype for better visualization.  

  1. While AOI is described in the legend for Figure 1, please also define AOI in the main text. AOI and ROI seem highly similar and are used somewhat interchangeably. A better definition of what each term is and what are the differences between the two would help the reader better understand the terminology.

            The description of AOI is added to in the main text (Line 166).            

  1. Line 419 should say (see “4.1.1. Data Normalization”).

            The number is now updated to “5.1.1.” (Line 532).

  1. Line 596, please define OME-TIFFs and MIAME/MINSEQE.

These acronyms are now defined in Lines 715-718 and in the List of Abbreviations (Lines 992-993). 

  1. Line 665, define MA plot

            The description of MA plot is added to Lines 787-788. 

  1. Line 699, define “PD”

Pharmacodynamic (PD) was defined in Line 271 and in the List of Abbreviations (Line 998).